# The Garlic Tree of Borneo, *Scorodocarpus borneensis* (Baill.) Becc. (Olacaceae): Potential Utilization in Pharmaceutical, Nutraceutical, and Functional Cosmetic Industries

**DOI:** 10.3390/molecules28155717

**Published:** 2023-07-28

**Authors:** Christophe Wiart, Afsana Amin Shorna, Mohammed Rahmatullah, Veeranoot Nissapatorn, Jaya Seelan Sathya Seelan, Homathevi Rahman, Nor Azizun Rusdi, Nazirah Mustaffa, Layane Elbehairy, Mazdida Sulaiman

**Affiliations:** 1Institute for Tropical Biology & Conservation, University Malaysia Sabah, Kota Kinabalu 88400, Malaysia; seelan80@ums.edu.my (J.S.S.S.); homa@ums.edu.my (H.R.); azizun@ums.edu.my (N.A.R.); m_nazirah@ums.edu.my (N.M.); 2Department of Biotechnology and Genetic Engineering, University of Development Alternative, Dhaka 1209, Bangladesh; afsanasharna2@gmail.com (A.A.S.); rahamatm@hotmail.com (M.R.); 3School of Allied Health Sciences, World Union for Herbal Drug Discovery (WUHeDD), and Research Excellence Center for Innovation and Health Products (RECIHP), Walailak University, Nakhon Si Thammarat 80160, Thailand; veeranoot.ni@wu.ac.th; 4Department of Chemistry, Faculty of Science, University Malaya, Kuala Lumpur 50603, Malaysia; lelbehairy@gmail.com (L.E.); mazdidas@um.edu.my (M.S.)

**Keywords:** garlic tree, nutraceutical, organosulfurs, *Scorodocapus borneensis*

## Abstract

*Scorodocarpus borneensis* (Baill.) Becc. is attracting increased attention as a potential commercial medicinal plant product in Southeast Asia. This review summarizes the current knowledge on the taxonomy, habitat, distribution, medicinal uses, natural products, pharmacology, toxicology, and potential utilization of *S. borneesis* in the pharmaceutical/nutraceutical/functional cosmetic industries. All data in this review were compiled from Google Scholar, PubMed, Science Direct, Web of Science, ChemSpider, PubChem, and a library search from 1866 to 2022. A total of 33 natural products have been identified, of which 11 were organosulfur compounds. The main organosulfur compound in the seeds is bis-(methylthiomethyl)disulfide, which inhibited the growth of a broad spectrum of bacteria and fungi, T-lymphoblastic leukemia cells, as well as platelet aggregation. Organic extracts evoked anti-microbial, cytotoxic, anti-free radical, and termiticidal effects. *S. borneensis* and its natural products have important and potentially patentable pharmacological properties. In particular, the seeds have the potential to be used as a source of food preservatives, antiseptics, or termiticides. However, there is a need to establish acute and chronic toxicity, to examine in vivo pharmacological effects and to perform clinical studies.

## 1. Introduction

Approximately 14,500 species of flowering plants are estimated to exist in Malaysia (including Sarawak and Sabah), of which 2500 are medicinal [1]. Out of these, four have captured global attention for their therapeutic potentials: *Calophyllum teysmannii* var *inophyloides* (King) P.F. Stevens from Sarawak as a source of antiretroviral agents [2]; *Mitragyna speciosa* Korth. for opioid addiction [3]; *Labisia pumila* Benth. & Hook. f. as a female aphrodisiac [4]; and *Eurycoma longifolia* Jack as a male aphrodisiac, with the latter probably being hazardous [5]. Rainforest medicinal plants in Malaysia and Southeast Asia are on the verge of extinction due to incessant burning and logging for palm oil [6]; in addition, there is a need to document and examine their potential utilization in pharmaceutical/nutraceutical/functional cosmetic applications before their extinction.

Infections resulting from *Pseudomonas aeruginosa*, carbapenem-resistant *Acinetobacter baumannii*, vancomycin-resistant *Enterococcus faecium*, and methicillin-resistant *Staphylococcus aureus* (MRSA) might soon become completely untreatable [7]. In 2009, a multidrug-resistant yeast *Candida uris* emerged in Japan with a death rate of approximately 50%. This yeast resists azoles, echinocandin, and amphotericin B, stays alive for seven days on inanimate surfaces, and evades ICU’s sanitation protocols [8]. If left unchecked, 10 million people could die yearly from untreatable microbial infections by 2050 [9]. The problem of microbial resistance is not limited to hospitals, as it also affects animal husbandry [10] and crops [11]. Concerning crops, insecticide resistance accounts for enormous losses and the dire perspective of food shortages [12]. The *Coptotermes curvignathus* and *Coptotermes gestroi* rubber termites’ infestation of fruit tree and rubber tree plantations in Southeast Asia requires the discovery of natural and environmentally friendly termiticides [13].

In this context, there is, now more than ever, a need to identify original molecules with antimicrobial and insecticidal properties, and these compounds could potentially be found in angiosperms, especially those used medicinally, which have evolved to survive phytopathogenic microbes and insects. There is also a globally growing interest in nutraceuticals, which now represent a significant portion of the current healthcare market. In 2020, the herbal supplement sales in the US reached a record-shattering USD 11,261 billion, representing an increase of approximately 18% from 2019. Garlic was among the top 10 products used for hypertension in 2020 [14], as well as to boost the immune system, while, in vitro, it inhibits the growth of lymphocytic leukemia cells on account of organosulfur compounds such as allicin [15]. For approximately the last two decades, researchers have been collecting ethnopharmacological data on the medicinal plants of Southeast Asia with the aim of developing antibacterial and other pharmacologically important leads for clinical practice, veterinary medicine, crop infections, cosmetics, and nutraceutical applications [16,17,18,19,20]. In Malaysia, one medicinal plant used for food and containing organosulfurs is *Scorodocarpus borneensis* (Baill.) Becc., which is beginning to become a center of interest as a potential source of health products in Southeast Asia. In this context, this review aims to summarize the current knowledge on the taxonomy, habitat, distribution, medicinal uses, natural products, pharmacological activity, toxicology, and potential utilization of *S. borneesis* in pharmaceutical/nutraceutical/functional cosmetic applications.

## 2. Methods

Databases such as PubMed, Medline, Science Direct, Thomson Reuters ISI Web of Science, ChemSpider and PubChem were searched, compiling the following keywords: “*Scorodocarpus borneensis*”, “garlic tree”, “traditional medicine”, “phytochemical composition of garlic tree”, and “herbal medicine”. In addition, the scientific literature available within the last two centuries (1866–2022) was considered in this review, resulting in the selection of 68 publications.

## 3. Results

### 3.1. Taxonomy, Habitat, Distribution, Ecology, and Botanical Description

*S. borneensis* is a massive timber tree belonging to the Olacaceae A. L. de Jussieu ex R. Brown (1818) family in the order Santalales R. Br. ex Bercht. & J. Presl (1820). It grows in the rainforests of Thailand, Malaysia, and Indonesia and reaches approximately 60 m in height. This plant has a unique and stout odor of garlic that can be perceived up to 100 m away. Its bark is dark brown, and flaky, and the inner bark is reddish-orange and sappy. The leaves are simple, alternate, and exstipulate. The petioles are 1.5–2 cm long. The blades are elliptic, glossy, 4–9 cm × 10–22 cm, dark green, cuneate at the base, acuminate at the apex, and marked with 5–6 pairs of secondary nerves. The racemes are axillary and approximately 4 cm long. The calyx is minute, tubular, and vaguely 4–5-lobed. The corolla is tubular, white, hairy, 4–5-lobed, and approximately 1 cm long. A total of 8–10 sessile stamens with filamentous anthers are present. The ovary is slender, hairy and approximately 5 mm long. The drupes are somewhat globose, green, and comprise a woody endocarp approximately 5 cm across (Figure 1) carved with blood vessel-like lines sheltering a pungent, spongy, and oily seed with the size and appearance of a somewhat light brown ping pong ball [21]. The transversal section of the germinating seeds presents purple patches (personal observation).

The Bornean garlic tree was first botanically examined by Oduardo Beccari in Mt. Matang (Sarawak) in May 1866 and a description in the Italian language under the name *Scorodocarpus borneensis* Becc. is available in the “*Nuovo Giornale Botanico Italiano*” volume 9 pages 274–279. Beccari noted a very strong garlic odor, “*fortissimo odore de aglio*”. Another early description is given by Henry Ernest Baillon, who calls the plant *Ximenia borneensis* Baill (page 271 of Adansonia volume 11 under the title *Receuil d’observations botaniques*).

### 3.2. Medicinal Uses

Henry Nicholas Ridley listed the plant in his “Malay Materia Medica” (J. Straits Medical Assn. 5, 122) under the local name “*kulim*” and wrote the following: “A large tree every part of which smells strongly of onions”. The fresh seeds are used for medicine in Malaysia and Indonesia. In Peninsular Malaysia, a decoction of seeds is ingested to prevent or treat kidney failure and fresh pounded seeds are applied to ringworms. Other uses for the seeds include high blood pressure, stroke, heart diseases, and food poisoning, as well as a substitute for garlic, from which the Malay name “*bawang hutan*” literally meaning garlic of the forest. In Sabah, the Murut people use the seeds for food (local name: *Sedau*) [22]. In Kalimantan, the tree is called “*kayu bawang*” meaning wood garlic and the fruits are eaten in place of garlic [23] and used prevent meat and oil from decay [24]. In Sumatra, the locals use the seeds for food purposes and for intestinal worms [25].

### 3.3. Antibacterial and Antifungal Activity of Extracts

There is a great need for affordable antibacterial agents to confront the emergence of resistant bacteria in clinical practice, sanitation, food preservation, veterinary medicine, and for the treatment of infected crops [7]. The seeds and leaves of *S. borneensis* contain antimicrobial principles (Table 1).

The methanol extract of leaves (60 µL of 50% *w*/*v* in 5 mm well) inhibited the growth of MRSA, *E. coli*, and *C. albicans* while delaying the bacterial decay of red tilapia filets [24].The petroleum ether extract of fresh seeds inhibited the growth of *B. cereus*, *P. aeruginosa*, *C. albicans*, and *A. ochraceus* (Table 2). 

A preparation made of 50 mg of this oil mixed with 1 g of paraffin was able to protect rodents against *Microsporium* sp. skin infection as well as ringworm [25]. Essential oil of leaves (yield 0.3%; 20 µL/5 mm well) in 6 mm agar wells of inhibited the growth of *S. sobrinus*, *S. nutans*, *C. albicans*, *S. aureus*, and *S.typhi* [51,52]. A dichloromethane extract of leaves inhibited the replication of the Hepatitis B virus [53]. Indonesian workers have attempted to identify active principles [54]. 

The ethyl acetate extract of bark (100 µL of a 10% *w*/*v*/6 mm well) inhibited the growth of *S. aureus* and *E. coli* [33]. The mechanism of actions of these extracts are not yet known.

**Table 2 molecules-28-05717-t002:** The pharmacological activities of extracts and secondary metabolites from *S. borneensis*.

Extract/Secondary Metabolites	Activities In Vitro	References
Petroleum ether extract of seeds	*Bacillus cereus*, IZD = 25 mm	[25]
	*Pseudomonas aeruginosa*, IZD = 50 mm	[25]
	*Candida albicans*, IZD = 19.2 mm	[25]
	*Aspergillus ochraceus*, IZD = 25 mm	[25]
*n*-hexane extract of seeds	Mouse lymphocytic leukemia cells, IC_50_ = 15.3 μg/mL	[33]
	DPPH, IC_50_ = 60 ppm	[55]
Methanol extract of seeds	DPPH, IC_50_ = 86.2 ppm	[55]
Ethanol extract of seeds	DPPH, IC_50_ = 14.5 ppm	[55]
*n*-Hexane extract of bark	*C. curvignathus*, LC_50_ = 0.01%	[13]
Ethyl acetate extract of bark	*C. curvignathus*, LC_50_ = 0.02%	[13]
	Brine shrimps, LC_50_ = 31.1 ppm	[34]
	DPPH, IC_50_ = 55.5 ppm	[24]
Methanol extract of bark	DPPH, IC_50_ = 52.4 ppm	[24]
Methanol extract from leaves	DPPH, IC_50_ = 36.8 ppm	[24]
Essential oil of leaves	DPPH, IC_50_ = 715.9 µg/mL	[52]
(**1**)	*S. aureus*, MIC = 12.5 μg/mL	[26,27,28,29]
	*Micrococcus luteus*, MIC = 25 μg/mL	[26,27,28,29]
	*Bacillus subtilis*, MIC = 12.5 μg/mL	[26,27,28,29]
	*Mycobacterium smegmatis*, MIC = 12.5 μg/mL	[26,27,28,29]
	*Escherichia coli*, MIC = 12.5 μg/mL	[26,27,28,29]
	*Candida albicans*, MIC = 25 μg/mL	[26,27,28,29]
	*Saccharomyces cerevisae*, MIC = 25 μg/mL	[26,27,28,29]
	*Mucor racemosus*, MIC = 12.5 μg/mL	[26,27,28,29]
	*Aspergillus niger*, MIC = 25 μg/mL	[26,27,28,29]
	Platelets aggregation, IC_50_ = 2.3 × 10^−4^ M	[32]
(**2**)	Platelets aggregation, IC_50_ = 2.9 × 10^−4^ M	[32]
(**3**)	*Staphylococcus aureus*, MIC = 50 μg/mL	[26,27,28,29]
	*Micrococcus luteus*, MIC = 50 μg/mL	[26,27,28,29]
	*Bacillus subtilis*, MIC = 12.5 μg/mL	[26,27,28,29]
	*Mycobacterium smegmatis*, MIC = 50 μg/mL	[26,27,28,29]
	*Escherichia coli*, MIC = 50 μg/mL	[26,27,28,29]
	*Candida albicans*, MIC = 50 μg/mL	[26,27,28,29]
	*Saccharomyces cerevisae*, MIC = 12.5 μg/mL	[26,27,28,29]
	*Mucor racemosus*, MIC = 25 μg/mL	[26,27,28,29]
	*Aspergillus niger*, MIC = 12.5 μg/mL	[26,27,28,29]
	*Bacillus cereus*, MM = 25 mg/disc	[25]
	*Pseudomonas aeruginosa*, MM = 25 mg/disc	[25]
	*Aspergillus ochraceus*, MM = 12.5 mg/disc	[25]
	*Saccharomyces lipolytica*, MM =12.5 mg/disc	[25]
	*Candida lipolytica*, MM = 12.5 mg/disc	[25]
	*Penicillium* sp., MM = 20 mg/disc	[25]
	*Acremonium* sp., MM = 1 mg/disc	[25]
	*Microsporium* sp., MM = 0.5 mg/disc	[25]
	*Pseudoscaellia boedes*, MM = 22 mg/disc	[25]
	Platelets aggregation, IC_50_ = 0.4 × 10^−4^ M	[32]
	T-Lymphoblastic leukemia cells, IC_50_ = 3 µg/mL	[25]
(**4**)	Platelets aggregation, IC_50_ = 1.2 × 10^−4^ M	[32]
(**7**)	T-Lymphoblastic leukemia cells, IC_50_ = 24 µg/mL	[25]
(**9**)	Mouse lymphocytic leukemia cells, IC_50_ = 1.1 µg/mL	[33]
	DPPH, IC_50_ = 51.1 ppm	[56]
(**11**)	Mouse lymphocytic leukemia cells, IC_50_ = 1.7 µg/mL	[56]
	DPPH, IC_50_ = 42.2 ppm	[56]
(**12**)	*Bacillus cereus*, IZD = 12 mm	[25]
	*Pseudomonas aeruginosa*, IZD = 11 mm	[25]
	T-Lymphoblastic leukemia cells, IC_50_ = 0.3–1 µg/mL	[25]
(**13**)	Brine shrimps, LC_50_ = 42.3 ppm	[51]

MIC: minimum inhibitory concentration (µg/mL); MIM: minimum inhibitory mass (g/disc); IZD: inhibition zone diameter (6 mm paper disc impregnated with 100 mg/mL solution); IC_50_: inhibitory concentration 50% (ppm); LC_50_: lethal concentration 50% (ppm).

### 3.4. Cytotoxicity and Brine Shrimp Toxicity of Extracts

Organosulfur compounds from garlic in vitro inhibited the growth of lymphocytic leukemia cells [15]. Likewise, extracts of *S. borneensis* are cytotoxic for leukemia cells (Table 2). The methanol extracts of seeds, leaves, and bark inhibited T-Lymphoblastic leukemia CEM-SS cells, while the *n*-hexane extract of seeds inhibited the growth of mouse lymphocytic leukemia L1210 cells [25]. In a subsequent study, ethyl acetate extract of bark was toxic for brine shrimps [23]. 

### 3.5. Termiticidal Activity of Extracts

*Coptotermes curvignathus* and *Coptotermes gestroi* account for significant losses in fruit trees, timber, coconut, rubber tree plantations and paddy fields in Southeast Asia, for which environmentally friendly termiticides are desperately needed. *n*-Hexane and ethyl acetate extract of bark were toxic for *C. curvignathus* with the LC_50_ values of 0.01 and 0.02% (*w*/*v*), respectively [13]. The acetone extract of wood exhibited repellent activity against *Coptotermes gestroi* [34].

### 3.6. Radical-Scavenging Activity of Extracts

The involvement of free radicals in the pathophysiology of cardiovascular, metabolic, and neurodegenerative diseases is well-established [57] and the chemo-preventive effect of garlic (*Allium sativum* L.) organosulfur is owed, at least in part, to radical-scavenging effects [58]. The methanol extract from leaves, bark, and seeds [24], ethyl acetate extract of bark [33], the ethanol extract of seeds [55], and the *n*-hexane extracts of seeds scavenged DPPH free radicals [34]. The essential oil of leaves displayed meek radical-scavenging activities [52,53] (Table 2). The anti-free radical activities of organosulfur compounds in this plant need to be examined. The organosulfur compounds in general can protect cells against free radicals by interacting with oxidative stressors and affecting the function of redox-sensitive cysteine proteins [58]. 

### 3.7. Organosulfur Compounds

The seeds of *S. borneensis* radiate an intense garlic odor due to the volatile organosulfur compounds (Figure 2, Table 1 and Table 2) first identified by Kubota and coworkers (1994) [26].

Methylthiomethyl(methylsulfonyl)methyl disulfide (**1**), methyl methylthiomethyl disulfide (**2**), and bis-(methylthiomethyl)disulfide (**3**). Methylthiomethyl(methylsulfonyl)methyl disulfide (**1**) has an odor threshold as low as 1.6 ppm and inhibited the growth of *S. aureus* (FDA 209P), *Micrococcus luteus* (PCI 1002), *Bacillus subtilis* (PCI 219), *Mycobacterium smegmatis* (ATCC 607), *Escherichia coli* (NIHJ), *Candida albicans* (KF 1), *Saccharomyces cerevisae* (ATCC 9763), *Mucor racemosus* (IFO 4581), and *Aspergillus niger* (KF 105), while being inactive for *P. aeruginosa* (IFO 3080) [26] (Table 2).

Bis-(methylthiomethyl) disulfide (**3**) inhibited the growth of *S. aureus* (FDA 209P), *Bacillus subtilis* (PCI 219), *Mycobacterium smegmatis* (ATCC 607), *Escherichia coli* (NIHJ), *Candida albicans* (KF 1), *Saccharomyces cerevisae* (ATCC 9763), *Mucor racemosus* (IFO 4581), and *Aspergillus niger* (KF 105), while being inactive for *M. luteus* (PCI 1002). In this experiment, methyl methylthiomethyl disulfide (**2**) was inactive against all the strains tested. Being a major constituent, bis-(methylthiomethyl) disulfide (**3**) has been suggested to be used as a flavoring agent and food preservative [26,27,28,29]. Subsequently, bis-(methylthiomethyl)disulfide (**3**) inhibited the growth of *Bacillus cereus*, *Pseudomonas aeruginosa*, *Aspergillus ochraceus*, *Saccharomyces lipolytica*, *Candida lipolytica*, and *Saccharomyces lypolitica*, *Penicillium* sp., *Acremonium* sp., *Microsporium* sp., and *Pseudoscaellia boedes* [25] (Table 2).

Lim et al., 1998 [29] further identified from the seeds 2,4,5,7-tetrathiaoctane 4,4-dioxide (**4**) and 5-thioxo-2,4,6-trithiaheptane 2,2-dioxide (**5**) both antibacterial and antifungal as well as *O*-ethyl-*S*-methylthiomethyl thiosulfite (**6**). These linear organosulfur compounds are not common in flowering plants. Bis(methylthiomethyl) disulfide (**3**) is only known to be produced by *Gallesia integrifolia* (Spreng.) Harms in the family Phytolaccaceae [30] in the order Caryophyllales Juss. ex Bercht. & J. Presl (1820), which is a neighbor to the order Santalales in the Clade Malvids. Organosulfur compounds different from those of *S. borneensis* are found in members of the genus *Allium* L. (family Amaryllidaceae, Clade Monocots). Lim et al., 1998 [29] proposed a biosynthetic pathway for *S. borneensis* organosulfur compounds similar to that of plants in the genus *Allium* L.; however, Kubota et al., 1998 [31] provided evidence for (*R*s)-3-[(methylthio)methylsulfinyl]-l-alanine and *S*-[(methylthio)methyl]-l-cysteine as the precursors of methyl methylthiomethyl disulfide (**2**) and bis(methylthiomethyl) disulfide (**3**), respectively. 

Thrombolytic agents are needed to prevent strokes, which are one of the major causes of death globally. Organosulfur compounds in the seeds of *S. borneensis* inhibit platelet aggregation (Table 1 and Table 2). Methylthiomethyl (methylsulfonyl)methyl disulfide (**1**) (2,4,5,7-tetrathiaoctane 2,2-dioxide), methyl methylthiomethyl disulfide (**2**) (2,4,5-trithiahexane I), bis-(methylthiomethyl) disulfide (2,4,5,7-tetrathiaoctane II) (**3**), and 2,4,5,7-tetrathiaoctane 4,4-dioxide (**4**) inhibited the aggregation of rabbit platelets induced by collagen [32]. Bis-(methylthiomethyl) disulfide (**3**) inhibited the growth of T-Lymphoblastic leukemia CEM-SS cells [25]. The antimicrobial and cytotoxic modes of action of these organosulfur compounds are unknown and could involve, at least in part, the disruption of DNA and cellular membranes [57]. 

### 3.8. Indole Alkaloids

The seeds of *S. borneensis*, especially when they germinate, yield long-chain and purple-colored indole alkaloids (Figure 2, Table 1 and Table 2) of a very uncommon constitution in flowering plants such as 13-docosenoyl serotonin (**7**), scorodocarpine A (**8**), B (**9**), and C (**10**) [25] as well as dehydroxy scorodocarpine B (**11**) [25,56,59]. These alkaloids tend to inhibit the growth of leukemia cells in vitro. 13-Docosenoyl serotonin (**7**) inhibited the growth of T-Lymphoblastic leukemia CEM-SS cells [25], while scorodocarpine B (**9**) and dehydroxyl scorodocarpine B (**11**) were cytotoxic towards L1210 mouse lymphocytic leukemia cells [33] and scavenged 2,2-diphenyl-1-picrylhydrazyl (DPPH) free radicals [56]. Being indole alkaloids with long chains, these rare alkaloids might have neurotrophic properties [60].

### 3.9. Sesquiterpenes

The seeds contain scodopin (**12**) and the bark cadalene-β-carboxylic acid (**13**) [34,60] (Figure 2, Table 1). Scodopin (**12**) inhibited the growth of *B. cereus* and *P. aeruginosa*, and was cytotoxic for T-Lymphoblastic leukemia CEM-SS cells [25]. Cadalene-β-carboxylic acid (**13**) is toxic to brine shrimps [34] (Table 2).

### 3.10. Megastigmanes

Grasshopper ketone (**14**), icariside B1 (**15**), blumenol B (**16**), and scorospiroside (**17**) have been identified from the leaves Figure 2, Table 1). Grasshopper ketone (**14**) decreased cytokine production by mice splenocytes challenged with in concanavalin A while at concentrations ranging from 0.1 to 100 µg/mL inhibited the production of nitric oxide, interleukin-6, interleukin-1β, and tumor necrosis factor-α by RAW 264.7 cells challenged with lipopolysaccharides. Grasshopper ketone (**14**) inhibited the growth of cress shoots at concentrations greater than 10 μmol/L. At 600 ppm, blumenol B (**16**) inhibited the growth of *Miscanthus floridulus* [35,36,37,61,62].

### 3.11. Flavonoid Glycosides

A phytochemical analysis of the leaves resulted in the identification of lucenin-2 (luteolin 6,8-di-C-glucoside) (**18**), vicenin-2 (apigenin 6,8-di-C-glucoside) (**19**), isoschaftoside (apigenin 6-C-arabinosyl-8-C-glucoside) (**20**), tricin 7-*O*-glucoside (**21**), and isorhamnetin 3-*O*-robinobioside (**22**) [62] (Figure 2). Lucenin-2 (**18**) inhibited the growth of *P. aeruginosa* (ATCC 27853), *E. coli* (ATCC 11229), and *K. pneumoniae* (ATCC 27736) with the MIC values of 8, 64, and 64 μg/mL, respectively [38]. Vicenin-2 (**19**) displayed antiglycation [39], anti-inflammatory [40], antiseptic [41], antiosteoporosis [42] properties, and proved to be of potential value against prostate cancer [43] and colon cancer [44]. Isoschaftoside (**20**) is phytotoxic [45], hepatoprotective [46], and diuretic [47]. Isorhamnetin 3-*O*-robinobioside (**22**) is antigenotoxic [48], and immunostimulant [49,50].

### 3.12. Miscellaneous

5,7-Dihydroxy-2-methylchromone-7-*O*-β-D-apiosyl (1,6)-β-D-glucoside (**23**) was identified from the leaves as well as uridine (**24**) *Threo*-guaiacylglycerol (**25**) and *erythro*-guaiacylglycerol (**26**) [62]. Kubota et al., 1994 [26] identified via GC-MS analysis (by comparison with standards) from the essential oil of the fresh seeds (89 mg/100 g) ethanal (96.4%) (**27**) and traces of methanethiol (**28**), dimethyl sulfide (**29**), propane thiol (**30**), dimethyl sulfide (**31**), (*E*)-2-hexanal (**32**), and 1.3 dithietane (**33**) (Figure 2). Using the passage of helium in fresh grated seeds, they identified bis-(methylthiomethyl)disulfide (**3**) (96.4%) via GC-MS as a main constituent, in addition to identifying methyl methylthiomethyl disulfide (**2**) (2.4%) and traces of tris(methylthio)methane. Other reports on the GC-MS analysis of hexane and ethyl acetate extract of stembark of *S. borneensis* are available [13,33,54] but without indication of the molecular mass of the various peaks observed and without reference compounds. 

### 3.13. Toxicity, Side Effects, and Drug Interaction

There are no preclinical studies available on the toxicity of the fruits of *S. borneensis.* Petroleum ether extract of seeds had an intraperitoneal lethal dose 50% (LD_50_) value of 275 mg/Kg in mice and, when mixed at 50 mg in 1 g of paraffin, it did not irritate the skin of rabbits [25]. The use of the seeds for food by the Malays, Indonesians, and other ethnic groups in North Borneo since the dawn of time might be an indication of a lack of toxicity when taken at a dietary dose; however, acute or chronic toxicity studies, including drug interaction studies are needed, especially regarding the thrombolytic activities of organosulfur compounds [32]. 

## 4. Conclusions

In conclusions, it is clear that *S. borneensis* extracts and natural products have important and patentable pharmacological properties, including antimicrobial, cytotoxic, anticoagulant, termiticidal, antioxidant, and phytotoxic activities. Of particular interest, organosulfur compounds inhibit the growth of pathogenic microbes, leukemic cells, and prevent the aggregation of platelets. In sum, *S. borneensis* and its natural products have potential utilization in pharmaceutical/nutraceutical/functional cosmetic applications. The fixed oil expressed from the seeds could be used as an active ingredient for food preservatives, antiseptics, or termiticides, and generate financial benefits for exportation in Southeast Asia. It is now up to local entrepreneurs and pharmaceutical, nutraceutical, and cosmetic companies to take the opportunity to develop and commercialize products from this plant. The yield of their oil is good and a tree produces hundreds of fruits, representing an economical material for exportation and for the development of local herbal businesses.

## Figures and Tables

**Figure 1 molecules-28-05717-f001:**
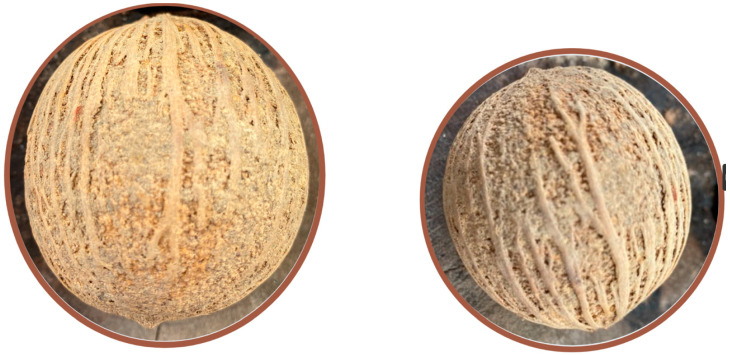
Endocarp of *S. borneensis*.

**Figure 2 molecules-28-05717-f002:**
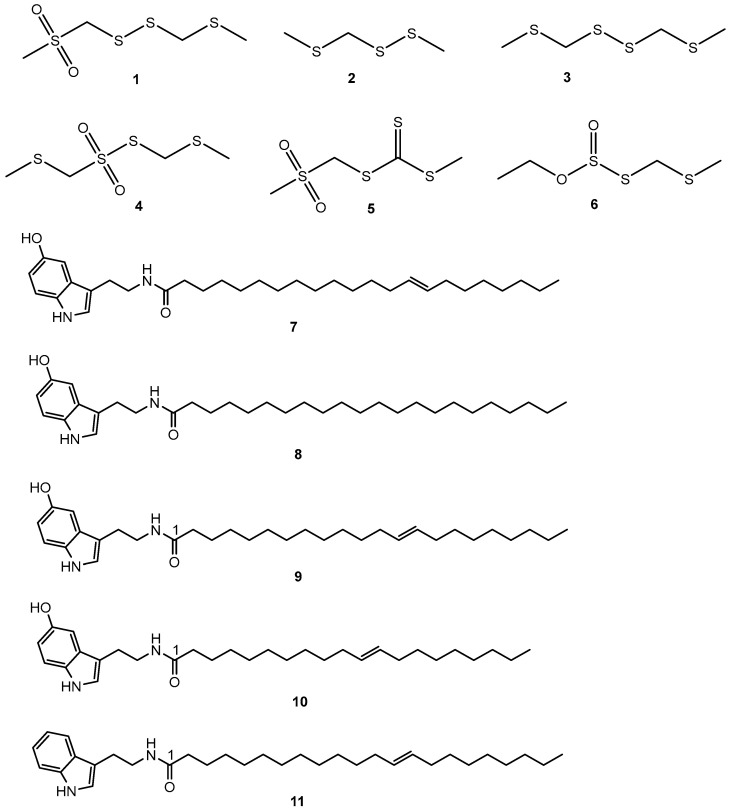
Secondary metabolites identified from *S. borneensis*.

**Table 1 molecules-28-05717-t001:** The pharmacological activities of secondary metabolites identified from *S. borneensis*.

SM	PP	Activities In Vitro	References
(**1**)	Seeds	Antibacterial, antifungal	[26,27,28,29,30,31]
		Platelet aggregation inhibitor	[32]
(**2**)	Seeds	Platelet aggregation inhibitor	[32]
(**3**)	Seeds	Antibacterial, antifungal	[25]
		Platelet aggregation inhibitor	[32]
		Cytotoxic	[25]
(**4**)	Seeds	Platelet aggregation inhibitor	[32]
(**7**)	Seeds	Cytotoxic	[15]
(**9**)	Seeds	Cytotoxic	[33]
(**11**)	Seeds	Cytotoxic	[33]
(**12**)	Seeds	Antibacterial	[25]
		Cytotoxic	[25]
(**13**)	Seeds	Toxic of *Artemia salina*	[34]
(**14**)	Leaves	Anti-inflammatory in vitro	[35]
	Leaves	Phytotoxic	[36]
(**16**)	Leaves	Phytotoxic	[37]
(**18**)	Leaves	Antibacterial	[38]
(**19**)	Leaves	Antiglycation	[39]
		Anti-inflammatory	[40]
		Antiseptic	[41]
		Antiosteoporosis	[42]
		Cytotoxic	[43,44]
(**20**)	Leaves	Phytotoxic	[45]
		Hepatoprotective	[46]
		Diuretic	[47]
(**22**)	Leaves	Antigenotoxic	[48]
		Immunostimulant	[49,50]

SM: Secondary metabolites; PP: Plant parts.

## Data Availability

Not applicable.

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
