# Peer review of "The Garlic Tree of Borneo, Scorodocarpus borneensis (Baill.) Becc. (Olacaceae): Potential Utilization in Pharmaceutical, Nutraceutical, and Functional Cosmetic Industries"

_molecules, 2023, doi:10.3390/molecules28155717_

Round 1

Reviewer 1 Report

This review reports an interesting and informative summary of what is currently know regarding the bioactivities of extracts and natural products derived from the Garlic tree S. borneensis.

Many of the reported bioactivities are quoted as long sentences of numbers (and microorganisisms) within the text (e.g. in section 3.3.1). Whilst informative the review could be made much more accessible to the reader if much of this information was instead presented in a tabular format. It would also be useful if in addition to the seeds an image of the garlic tree and its leaves were also included in the introduction to enable the reader to visualise the plant species being discussed.

Author Response

Answer: Thank you for the comment. We have amended accordingly. We do not have photos of the leaves

Reviewer 2 Report

Authors have reported the work on “The Garlic Tree of Borneo, Scorodocarpus borneensis (Baill.) 2 Becc. (Olacacaeae)”. Authors have reviewed and compiled using the resources available on Google Scholar, PubMed, Science Direct, 23 Web of Science, ChemSpider, PubChem, and library search from 1866 to 2022. The compilation is exhaustive, to the point, specific, and concerning the effect of phytochemicals on the antimicrobial, cytotoxic, anti-free radical, and termiticidal effects. The possibility of seeds having the potential to be used as a source of food preservatives, antiseptics, or termiticides has been proposed.

Although the review is well written, including the following may further improve the manuscript:

1.      The mechanism of Antimicrobial action, Termiticidal role, and Toxicity effects against the quoted facts may be included.

Reference number 51 has capital letters in the paper title, which may be corrected.

Author Response

Answer: Thank you for the comment. To the best of our knowledge, there are no available reports on the mechanisms of action. We also have corrected the reference.

Reviewer 3 Report

  1. Title. Please reformulate the title to better figure out the goal/aims of this current review. It is not unclear what the authors want to discuss on the Garlic tree of Borneo.
  2. Abstract. Please state clearly what the aims of this current review. I would suggest to be specified the potential utilization the Garlic tree of Borneo in pharmacy/nutraceutical/ functional cosmetics.
  3. Keyword. Please write the keyword in alphabetical order.
  4. Introduction.

-       Please confirm and write the rationale behind why the authors choose to elaborate the Garlic tree of Borneo.

-       Please add more relevant studies.

-       What makes this current review differ from the previous study? Please explain in detail to show the novelty of this current review.

  1. Methods.

-       Please write the references that have been referred in the method section.

  1. Results and discussion

-       Please synchronize this current review aims with the discussion.

-       Please complete the label of figure 1 to show each of the picture.

-       Please more discuss the results instead of listing by comparing the results of this current work with other relevant studies.

-       I would suggest to re-organize the flow of the results section. I would suggest to discuss the natural product compounds in the Garlic Tree of Borneo, the continue to elaborate the group of compounds which are potential to be used in pharmaceutical/medicinal.

  1. Conclusion

-       Please add the future prospect of this current work.

Author Response

Title. Please reformulate the title to better figure out the goal/aims of this current review. It is not unclear what the authors want to discuss on the Garlic tree of Borneo.

Answer: Thank you for the comment. We have amended it accordingly.

Abstract. Please state clearly what the aims of this current review. I would suggest to be specified the potential utilization the Garlic tree of Borneo in pharmacy/nutraceutical/ functional cosmetics.

Answer: Thank you for the comment. We have amended it accordingly.

Keyword. Please write the keyword in alphabetical order.

Answer: Thank you for the comment. We have amended it accordingly.

Introduction.Please confirm and write the rationale behind why the authors choose to elaborate the Garlic tree of Borneo.

Answer: Thank you for the comment. We have amended it accordingly.

Please add more relevant studies.

Answer: Thank you for the comment. To the best of our knowledge, we have covered all existing studies

What makes this current review differ from the previous study? Please explain in detail to show the novelty of this current review.

Answer: The review of Kustiawan et al., 2021 is very limited and does not fulfill basic professional academic standards

[Kustiawan, P.M., Siregar, K.A.A.K., Saleh, L.O., Batistuta, M.A. and Setiawan, I.M., 2021. A Review of Botanical Characteristics, Chemical Composition, Pharmacological Activity and Use of Scorodocarpus borneensis. Biointerface Res. Appl. Chem12(6), pp.8324-8334.]

See for instance when describing the medicinal use only 1 line is written “Scorodocarpus borneensis traditionally used by forest villagers as spices and medicinal plants [5–7]”

Nothing on correct taxonomy, history, local names, distribution, no personal observation

The chemical constituents section is made of about 10 lines and reports compounds from some databases including progesterone and benzophenone without consulting the most important papers. The chemical structures are not correct and out of topic They did not provide a proper presentation of the natural products and to summarise this review is very mediocre and incomplete (not published in a peer reviewed journal nor Scopus indexed. One can find “etyhl acetate”.

 Methods.

-       Please write the references that have been referred in the method section.

Thank you for the comment. We have amended it accordingly.

 Results and discussion

Please synchronize this current review aims with the discussion.

Thank you for the comment. We have amended it accordingly.

Please complete the label of figure 1 to show each of the picture.

Thank you for the comment. We have amended accordingly in the text.

 Please more discuss the results instead of listing by comparing the results of this current work with other relevant studies.

Thank you for the comment. We have amended it accordingly.

 I would suggest to re-organize the flow of the results section. I would suggest to discuss the natural product compounds in the Garlic Tree of Borneo, the continue to elaborate the group of compounds which are potential to be used in pharmaceutical/medicinal.

Thank you for the comment. We have amended it accordingly.

Conclusion. Please add the future prospect of this current work.

Thank you for the comment. We have amended it accordingly.